# Ginseng Sprouts Attenuate Mortality and Systemic Inflammation by Modulating TLR4/NF-κB Signaling in an LPS-Induced Mouse Model of Sepsis

**DOI:** 10.3390/ijms24021583

**Published:** 2023-01-13

**Authors:** Seung-Ju Hwang, Jing-Hua Wang, Jin-Seok Lee, Ji-Yun Kang, Dong-Cheol Baek, Geon-Ho Kim, Yo-Chan Ahn, Chang-Gue Son

**Affiliations:** 1Liver and Immunology Research Center, Daejeon Oriental Hospital of Daejeon University, Daedukdae-ro 176 bun-gil 75, Seo-gu, Daejeon 35235, Republic of Korea; 2Institute of Bioscience & Integrative Medicine, Daejeon University, Daedukdae-ro 176 bun-gil 75, Seo-gu, Daejeon 35235, Republic of Korea; 3Department of Korean Medicine, Korean Medical College of Daejeon University, Daehak-ro 62, Dong-gu, Daejeon 34520, Republic of Korea; 4Department of Health Service Management, Daejeon University, Daehak-ro 62, Dong-gu, Daejeon 34520, Republic of Korea

**Keywords:** sepsis, ginseng sprout, ginsenoside Rd, inflammation, TLR4/NF-κB pathway, multi-organ failure

## Abstract

Sepsis leads to multi-organ failure due to aggressive systemic inflammation, which is one of the main causes of death clinically. This study aimed to evaluate whether ginseng sprout extracts (GSE) can rescue sepsis and explore its underlying mechanisms. C57BL/6J male mice (*n* = 15/group) were pre-administered with GSE (25, 50, and 100 mg/kg, *p.o*) for 5 days, and a single injection of lipopolysaccharide (LPS, 30 mg/kg, *i.p*) was administered to construct a sepsis model. Additionally, RAW264.7 cells were treated with LPS with/without GSE/its main components (Rd and Re) to explain the mechanisms corresponding to the animal-derived effects. LPS injection led to the death of all mice within 38 h, while GSE pretreatment delayed the time to death. GSE pretreatment also notably ameliorated LPS-induced systemic inflammation such as histological destruction in both the lung and liver, along with reductions in inflammatory cytokines, such as TNF-α, IL-6, and IL-1β, in both tissues and serum. Additionally, GSE markedly diminished the drastic secretion of nitric oxide (NO) by suppressing the expression levels of inducible nitric oxide synthase (iNOS) and cyclooxygenase 2 (COX2) in both tissues. Similar changes in TNF-α, IL-1β, NO, iNOS, and COX2 were observed in LPS-stimulated RAW264.7 cells, and protein expression data and nuclear translocation assays suggested GSE could modulate LPS-binding protein (LBP), Toll-like receptor 4 (TLR4), and NF-κB. Ginsenoside Rd could be a major active component in GSE that produces the anti-sepsis effects. Our data support that ginseng sprouts could be used as an herbal resource to reduce the risk of sepsis. The corresponding mechanisms may involve TLR4/NF-κB signaling and a potentially active component.

## 1. Introduction

Sepsis is a life-threatening condition, defined in the Third International Consensus as “organ dysfunction caused by a dysregulated host response to infection” [1]. Approximately 49 million people suffer from sepsis every year worldwide, and its fatality rate is estimated to be approximately 20% [2]. Sepsis does not allow appropriate blood flow to tissues due to a systemic inflammatory response and blood coagulation, causing multi-organ failure (MOF) and eventually leading to death. In the case of septic shock, which is characterized by persistent hypotension and dysregulated tissue perfusion, the in-hospital mortality rate is for 38% in Europe and North America [3]. The costs of medical care for sepsis are estimated to exceed USD 17 billion annually in the USA [4].

Sepsis processes include the initial overactivation of the immune system and consequential leukopenia, followed by a decrease in vascular resistance leading to hypotension [5]. Multiple subsets of immune cells, such as monocytes/macrophages, neutrophils, dendritic cells, and natural killer cells, contribute to pathophysiologic processes in sepsis [6]. Among them, macrophages play an important role throughout all stages of sepsis progression, mainly via the oversecretion of inflammatory cytokines, such as tumor necrosis factor-α (TNF-α), interleukin (IL)-6, and IL-1β, as well as other inflammatory mediators, including nitric oxide (NO) and prostaglandins, which ultimately aggravate systemic injury, producing immune collapse [7]. This macrophage-derived overactivation of immune reactions involves signaling of toll-like receptor 4 (TLR4) and nuclear factor-κB (NF-κB) translocation after recognizing pathogen-associated molecular patterns (PAMPs), likely lipopolysaccharide (LPS) [8]. Accordingly, modulation of the TLR4/NF-κB signaling pathway has been a therapeutic target for sepsis, and several candidates, such as eritoran and TAK-242, which are inhibitors of TLR4, have shown partial efficacy in both animal studies [9,10] and clinical trials [11,12].

*Panax ginseng*, a medicinal plant, has been used to nourish tonics and treat various health issues in Asian countries, including Korea and China, for thousands of years [13]. *P. ginseng* contains over 80 kinds of ginsenosides as its known major pharmacological compounds, as well as phytosterols, polyphenolic compounds, acidic polysaccharides, etc. [14]. To date, numerous studies have revealed various beneficial effects of *P. ginseng* and its derivatives, including ginsenosides, for the manipulation of fatigue [15], diabetes [16], chronic pain [17], and systemic inflammation [18]. 

Ginseng sprout (GS), of early harvested ginseng grown for weeks to months in a hydroponic or artificial soil cultivation system, bring benefits such as a shorter incubation period and lower cultivation cost [19]. GSs have recently been reported to contain mainly ginsenoside Rd of the protopanaxadiol type and Re of the protopanaxatriol type [20]. In particular, ginsenoside Rd was reported to have anti-inflammatory activity by suppressing NF-κB signaling [21]. Due to these economic and biological effects, GS has been widely used as a medicinal herb and edible plant in esaten Asia. Consequently, we hypothesized that GS water extract (GSE) might be a prospective candidate for treating sepsis.

This study, therefore, aimed to evaluate and explore the antisepsis effects and underlying mechanisms of GSE using an LPS-induced sepsis animal model and in vitro experiments.

## 2. Results

### 2.1. Fingerprinting Analysis of GSE

From the HPLC-based fingerprint and compositional analyses, ginsenosides Re and Rd were detected in GSE at retention times of 6.23, and 35.10 min, respectively, as expected. Semiquantitative analysis showed 47.88 mg/g Re and 13.26 mg/g Rd in GSE (Figure 1B–D).

### 2.2. GSE Delayed the Time to Septic Death

As expected, the peritoneal injection of LPS (30 mg/kg) killed all mice within 38 h of injection, while administration of GSE significantly delayed the time points to death in a dose-dependent manner (*p* < 0.05 or 0.01 for all doses, Figure 2A). Dexamethasone (5 mg/kg) saved 50% of the mice from death 72 h after injection.

### 2.3. GSE Ameliorated Systemic Inflammation

LPS (30 mg/kg) injection enlarged the spleen, caused the lung and liver to look pale (Figure 2B,C), and radically raised serum concentrations of TNF-α (approximately 97-fold), IL-6 (3.82-fold), and IL-1β (1.7-fold) at 12 h post-injection (*p* < 0.01, all the parameters, Figure 2D). Notably, administration of GSE significantly attenuated the alterations in spleen weight (*p* < 0.05 at 100 mg/kg) and levels of TNF-α, IL-6 (*p* < 0.05 or *p* < 0.01 at 50 and 100 mg/kg, respectively) and IL-1β (*p* < 0.05 at 100 mg/kg) (Figure 2C,D). In addition, LPS-induced depletion of total WBCs and lymphocytes, but the increase in monocytes in the peripheral blood was significantly attenuated by pre-treatment with GSE (*p* < 0.05 at 100 mg/kg, Table 1). Dexamethasone showed effects comparable with mg/kg GSE on spleen weight, IL-1β (but much significantly on TNF-α), and the depletion of white cells in the peripheral blood (but the increase in granulocytes was different from that observed after treatment with GSE).

### 2.4. GSE Attenuated Pulmonary and Hepatic Inflammation 

LPS (30 mg/kg) injection induced prominent inflammation in the lungs with a drastically destroyed alveolar structure and severe infiltration of inflammatory cells, whereas this effect was significantly attenuated by treatment with 100 mg/kg GSE. Similarly, GSE treatment (100 mg/kg) significantly attenuated LPS-induced liver inflammation as evidenced by elevated hepatic enzymes and infiltration of inflammatory cells, especially around the central venule (*p* < 0.05, Figure 3A–E). These beneficial effects of GSE were supported by the reduced levels of TNF-α, IL-6, and IL-1 β in both pulmonary and hepatic tissues (*p* < 0.05 or *p* < 0.01 at 100 mg/kg, Figure 3F,G). Similarly, 5 mg/kg of dexamethasone showed anti-inflammatory effects that were slightly stronger than those of 100 mg/kg GSE (*p* < 0.05 or 0.01, Figure 3). 

### 2.5. GSE Suppressed Oxidative-Stress-Related Molecules in Lung and Liver Tissues

LPS (30 mg/kg) injection substantially upregulated the expression of iNOS and COX2, as well as the production of NO in both the lung and liver. GSE treatments significantly reduced the oxidative molecules iNOS (*p* < 0.01) and COX2 (*p* < 0.05 for GSE50 and *p* < 0.01 for GSE100) and the production of NO (*p* < 0.01) in the lung (Figure 4A,B,E). Furthermore, GSE (especially, 50 and 100 mg/kg) downregulated the protein expression of iNOS (*p* < 0.01) and COX2 (*p* < 0.05) and the production of NO (*p* < 0.05 for GSE25, *p* < 0.01 for GSE 50 and 100) in the liver (Figure 4C,D,F).

### 2.6. GSE Modulated the TLR4/NF-κB Pathway in LPS-Stimulated RAW264.7 Cells

LPS-stimulated RAW264.7 cells exhibited activation of the TLR4/NF-κB signaling pathway, as shown by increased LBP, TLR4, nuclear translocated NF-κB, phosphorylated IκB, iNOS, and COX2 protein expression, while these alterations were significantly attenuated by GSE treatment (*p* < 0.05 or *p* < 0.01, Figure 5A–D). Furthermore, GSE treatments significantly inhibited the oversecretion of NO (*p* < 0.01 for all doses), TNF-α (*p* < 0.05 for GSE100 and *p* < 0.01 for GSE 200), IL-6 (*p* < 0.01 for all doses), and IL-1β (*p* < 0.01 for all doses) in LPS-stimulated RAW264.7 cells (Figure 5E,F). On the other hand, dexamethasone showed similar effects to GSE, except for a slightly different pattern in LBP and TLR4 expression (Figure 5).

### 2.7. Ginsenoside Rd Inhibited Inflammatory Cytokines and Mediators by Suppressing the Nuclear Translocation of NF-κB in LPS-Stimulated RAW264.7 Cells

Ginsenoside Rd and Re showed nontoxicity up to 50 and 100 μM, respectively (Figure 6A). Interestingly, low concentrations of Rd (12.5 and 25 μM) significantly increased cell proliferation. (Figure 6A and Appendix A). Rd (50 μM) treatment significantly inhibited the production of NO, and the expression of TNF-α, IL-6, and IL-1β in LPS-stimulated RAW264.7 cells compared to the control group (*p* < 0.01 for all parameters). Additionally, these anti-inflammatory effects were observed to a smaller extent after treatment (Figure 6B,C). Rd dominantly (and Re slightly) suppressed the nuclear translocation of NF-κB compared to the control group (*p* < 0.01) (Figure 6D,E). 

## 3. Discussion

To investigate the anti-sepsis potential of GSE and its underlying mechanisms, we applied LPS-stimulation in vivo and in vitro. LPS, a typical endotoxin, is used to induce the overactivation of immune cells and systemic inflammation to mimic various inflammation-related diseases [22,23]. Animal models of sepsis are commonly constructed by intraperitoneal injection of LPS since this model is similar to the pathophysiology of severe human sepsis [24]. As expected, a single dose of LPS (30 mg/kg) via peritoneal injection killed all mice within 38 h (Figure 2A). Septic death is generally caused by severe systemic inflammation leading to overall organ/tissue-failure including the lung, liver, and others, as shown in our results (Figure 2B and Figure 3A–E, and Table 1).

In our sepsis model, the serum levels of three major inflammatory cytokines, TNF-α, IL-6, and IL-1β, were drastically elevated at 12 h of induction (Figure 2D). In particular, TNF-α increased by approximately 100-fold, which is regarded as a core player in the septic inflammatory response because of its early and deterministic role in the amplified release of other cytokines, similar to the so-called cytokine storm [25]. IL-6 release generally follows that of TNF-α and is also an important inflammatory cytokine; several studies have shown that higher plasma levels of IL-6 are more likely to cause sepsis patients to die faster [26]. Thus, the serum levels of TNF-α and IL-6 have been proposed as prognostic parameters for patients suffering septic shock [27,28], and immunotherapy using anti-TNF-α or anti-IL-6 monoclonal antibodies, such as afelimomab and tocilizumab, have reduced overall mortality from severe sepsis [29,30]. In the current study, GSE pre-treatment significantly attenuated the production of TNF-α and IL-6 and another proinflammatory cytokine, IL-1β, in a dose-dependent manner in both serum and tissues (lung and liver) (Figure 2D and Figure 3F,G). These three cytokines are produced by all-around immune cells, predominantly macrophages and lymphocytes, which synergistically leads to septic conditions with other inflammatory mediators such as NO and prostaglandin E2 [7]. Based on the central contribution of monocyte-/macrophage-derived proinflammatory cytokines in fulminant inflammation and sepsis, modulating the action of macrophages is regarded as a key strategy to treat sepsis [31]. We confirmed that GSE inhibited the secretion of TNF-α, IL-6, and IL-1β in LPS-stimulated macrophages, using murine RAW264.7 cells (Figure 5F). These results support the positive effects of GSE on survival under septic conditions (Figure 2A).

In general, severe cases of sepsis progress into septic shock, a culprit of MOF, hypotension, and systemic coagulation, finally leading to death [32]. In particular, MOF is highly correlated with septic death. In the intensive care unit, patients without organ dysfunction had a 6% mortality rate, whereas those with four or more organ failures had a 65% rate of death [33]. In the current study, GSE treatment showed protective effects against MOF-like histopathological features in lung and liver tissues (Figure 3A–E). Although the mechanisms leading multiple organs to enter failure status during sepsis remain unclear, no doubt exists as to the overwhelming production of inflammatory cytokines along with oxidative stressors, including NO [34]. In our study, the excessive production of NO (in lung and liver tissue and RAW264.7 cells) was significantly attenuated by GSE (Figure 4E,F and Figure 5E). The effect of GSE on NO production was supported by the suppressed protein expression of iNOS in lung/liver tissues and macrophages (RAW264.7) (Figure 4A–D and Figure 5A,D). As an important contributor to sepsis pathogenesis, NO is involved in the induction of vascular relapse and hypotension, which inhibit the provision of oxygen and nutrients from the blood to multiple organs [35]. One phase 2 clinical trial showed that L-NMMA, an inhibitor of NOS, improved systemic vascular relapse, hypotension, and even mortality rates in patients with septic shock [36,37]. Additionally, leukopenia is a hallmark for the increased risk of MOF and death compared with leukocytosis [38]. From our data (Table 1), lymphocytopenia was dominantly accrued by sepsis-induced apoptosis [39] and significantly attenuated by GSE (200 and 400 mg/kg).

As a classic pathophysiology of sepsis development, the LPS-induced activation of the TLR4/NF-κB signaling pathway has been well defined [40]. TLR4 recognizes very small levels of circulating LPS bound by LBP and initiates a cascade that leads to the nuclear translocation of NF-κB, finally altering the gene expression of proinflammatory cytokines and mediators [41]. TLR4 knockout mice displayed notably improved survival rates and cardiac function under LPS-induced septic conditions [42], and TLR4 inhibitors including eritoran and TAK-242 have shown partial efficacy in both animal studies [9,10] and clinical trials [11,12]. As expected, GSE treatment notably suppressed TLR4/LBP expression in LPS-stimulated RAW264.7 cells (Figure 5A,B). In addition, GSE significantly suppressed both the LPS-induced nuclear translocation of NF-κB (after separation from IκB) and the overexpressions of proinflammatory cytokines (TNF-α, IL-6, and IL-1β) and iNOS and COX2 (Figure 5A,C,D). These results showed that GSE could diminish the severe inflammatory response by downregulating the TLR4/NF-κB signaling pathway.

Ginseng sprouts are new promising medicinal resources that have been highlighted due to their economic and pharmacological benefits [43]. GS is composed of various ginsenosides, including Rd, Re, and Rg [44]. In our results, GSE particularly contained Rd and Re at high concentrations of 13.26 and 47.88 mg/g in the extracts (Figure 1B–D), which corresponds approximately 2.2- and 15.0-fold to those in Korean red ginseng (data not shown). We also found that Rd strongly exerted the above anti-inflammatory effects including suppression NF-κB nuclear translocation (Figure 6D,E). In the present results, similar anti-inflammatory effects were found both in 200 μg/mL of GSE and 50 μM of ginsenoside Rd in LPS-stimulated RAW264.7 cells (Figure 5 and Figure 6). Therefore, the results indicated that the effect of GSE might be due to ginsenoside Rd, not Re. Several other animal studies have reported that Rd has strong anti-inflammatory activity by suppressing NF-κB signaling [21,45,46]. There are some similar data related to the TLR4/NF-κB pathway-mediated anti-inflammatory effects of herbal plants. For example, berberine, an active compound in *Coptidis Rhizoma* (a representative anti-inflammatory herb), showed anti-inflammatory effects via the modulation of the NF-κB pathway in animal models of sepsis [47], acute respiratory distress syndrome [48], and ulcerative colitis [49].

## 4. Materials and Methods

### 4.1. Chemicals and Reagents

The following reagents and chemicals were obtained from Sigma-Aldrich (Saint Louis, MO, USA): lipopolysaccharide from Escherichia coli serotype O111:B4, dexamethasone, Tris-base, sodium chloride, Triton X-100, 10% neutral formalin, calcium carbonate, calcium sulfate, and aqueous mounting buffer.

Other reagents and chemicals were purchased from the following manufacturers: ginsenoside Re, Rd, and Tween 20 (Glentham Life Science, Corsham, UK); skim milk (XOGENE, Daejeon, Korea); bovine serum albumin (GenDEPOT, Katy, TX, USA); hydrogen chloride, methanol, ethanol, and hydrogen peroxide (DUKSAN, Seoul, Korea); EZ-cytox (DoGenBio, Seoul, Korea); RIPA buffer (LPS solution, Daejeon, Korea); and Mayer’s hematoxylin (Wako Pure Chemical Industries, Osaka, Japan). 

### 4.2. Preparation of Ginseng Sprout Extract

GSs were obtained from the *Lim Sung-Man Ginseng Sprouts Laboratory* of *Samdaon* Corporation (Daejeon, South Korea) and washed with distilled water. Next, GSs were separated into aboveground parts (leaves and stems) and underground parts (roots) and then extracted with boiling water for two hours. Ginseng sprout extracts (GSE) were lyophilized by freeze-drying (−80 °C, 72 h) with eventual yields of 4.04%, and 3.88% (aboveground and underground, respectively) (Figure 1A). The acquired powders were stored at −20 °C for future use, and a voucher specimen (number 202003) was prepared. 

### 4.3. HPLC Fingerprinting Analysis

GSE was dissolved in 50% methanol and analysed with an Agilent 1260 high-performance liquid chromatography (HPLC) system (Agilent Technologies, Wilmington, DE, USA), which consisted of a binary pump, degasser, autosampler, column oven, and diode array detector for detection at a wavelength of 300 nm. An Agilent Eclipse XDB-C18 column (4.6 × 250 mm^2^, 5 μm) was used at room temperature (RT). The samples were eluted using a gradient of H_2_O and CH_3_CN increasing from 20% to 100% CH_3_CN over 45 min at a flow rate of 1 mL/min. Ginsenosides Re and Rd (Glentham Life Sciences, Wiltshire, UK) were used as internal references for the GSE. Finally, chromatograms were obtained with Agilent Chem Station software (Sever version: A.02.02(1.3.4)).

### 4.4. Animal Experiments and Ethics Approval 

A total of eighty C57BL/6J male mice (8 weeks old, 22–24 g) were purchased from Daehanbio-link (Eumseong-gun, Chung-Buk, Korea). All mice were maintained at room temperature (22 ± 2 °C) and 60 ± 5% relative humidity under a 12 h light:12 h dark cycle. They were given free access to tap water and a commercial pellet diet (Daehanbio-link).

After acclimatization for 1 week, the mice were randomly divided into two experimental sets. The first set (*n* = 30) was used to obtain tissues (spleen, lung, and liver) and blood samples, and the second set (*n* = 50) was used to determine the survival rate after injection of LPS from E. coli serotype O111:B4. The first set contained six groups (Nor, Con, GSE25, GSE50, GSE100, and Dexa5, *n* = 5/group), and the second set consisted of five groups (Con, GSE25, GSE50, GSE100, and Dexa5, *n* = 10/group). GSE (25, 50, and 100 mg/kg, dissolved in distilled water) were orally administered to the appropriate group twice a day for 5 continuous days, and the other groups (Nor, Con and Dexa5) were treated with distilled water. Body weights were determined on the first day and final day. There were no significant changes in body weight gain among the groups (Appendix A). On the final day of the experiment, the Dexa5 group was intraperitoneally injected with dexamethasone (5 mg/kg), and the other groups were injected with normal saline. One hour after the injection of dexamethasone or distilled water, mice in all sets, except those in the NOR group, were intraperitoneally injected with LPS (30 mg/kg). LPS doses were selected by a pilot experiment (Appendix A). The experimental designs are briefly summarized in Appendix A.

For the first set of experiments, mice were euthanized in a CO_2_ chamber (Jeungdo Bio & Plant, Seoul, Korea) 12 h after LPS injection, and whole blood was collected from the abdominal aorta. Immediately, a complete blood count (CBC) test was performed using an Exigo EOS analyzer (Boule Medical AB, Spanga, Sweden). Tissues (spleen, lung, and liver) were stored immediately. The survival rate was recorded at 1 h intervals from the start time to 72 h.

The animal study protocol was approved by the Institutional Animal Care and Use Committee of Daejeon University (Daejeon, Republic of Korea; approval no. DJUARB2022-001) and was conducted according to the Guide for the Care and Use of Laboratory Animals, published by the National Institutes of Health (NIH, MD).

### 4.5. Cell Culture and Chemical Treatment

RAW264.7 macrophages were obtained from the Korea Cell Line Bank (Seoul, Korea). The cells were maintained at 37 °C in high-glucose DMEM supplemented with 10% fetal bovine serum and antibiotics (penicillin and streptomycin sulfate, 100 units/mL and 100 μg/mL, respectively) in a humidified atmosphere with 5% CO_2_. Cells were incubated with GSE (50, 100, and 200 μg/mL) for 2 h, and then LPS was added at a final concentration of 100 ng/mL for 24 h. The experimental procedures are briefly summarized in Appendix A. GSE did not show cytotoxicity up to a concentration of 200 μg/mL (Appendix A).

### 4.6. Cell Viability

To determine cell viability, 20 μL of Ez-cytox reagents was added to cell and culture media in a 96-well plate for 2 h of incubation. Then, 100 μL of media was transferred into a new plate, and the absorbance at 450 nm was measured by using a spectrophotometer. Cell viability was calculated as a percentage of viable cells by comparison with the vehicle-treated control. 

### 4.7. Determination of NO

The NO that had accumulated in the lung, liver homogenates, and RAW264.7 cell culture medium was measured with the Griess reaction. Briefly, 40 or 100 μL of tissue homogenates and cell culture medium were mixed with 160 or 100 μL of Griess reagent and then incubated at 37 °C for 15 min. The absorbance was then measured at 540 nm with a spectrophotometer.

### 4.8. Determination of Proinflammatory Cytokines

According to the manufacturer’s instructions, the levels of inflammatory cytokines in serum, lung, liver, and culture media were measured using commercial enzyme-linked immunosorbent assay (ELISA) kits for TNF-α (BD Biosciences, San Jose, CA, USA), IL-6 (R&D Systems, Minneapolis, MN, USA) and IL-1β (R&D Systems, Minneapolis, MN, USA). The protein concentration was quantified with a Pierce BCA Protein Assay Kit (Thermo Fisher Scientific, Lafayette, CO, USA). The ELISA results from tissue samples were normalized according to the protein content (pg/mg protein).

### 4.9. Histopathological Analysis

For the histopathological evaluation, frozen lung and liver were sectioned at 8 μm and stained with Mayer’s hematoxylin and eosin (H&E). The stained samples were mounted on silane-coated slides using Aqueous-Mount (Scytek Laboratories, Logan, UT, USA). Five photographs per sample were acquired using an optical microscope (Leica, Wetzlar, Germany) at 200× magnification. For the objective evaluation of the histopathological findings, lung and liver injury scores were used to estimate the grade of severity. Lung inflammation scores were given in increments of two up to eight points after the examination of the thickness of the alveoli, the presence or absence of protein debris in the airspace, the infiltration of immune cells into interstitial space, and alveolar space. Liver inflammation was scored up to five points depending on the severity of immune cell infiltration and venous endothelial cell damage. 

### 4.10. Western Blot Analysis

Lung and liver tissue and cells were homogenized in PRO-PREP™ Protein Extraction Solution (iNtRON Biotechnology, Seoul, Korea). The supernatant was acquired; total protein concentrations were measured using a BCA Kit (Thermo Fisher Scientific, San Jose, CA, USA), and Laemmli’s sample buffer was added to adjust it to a certain protein concentration (1 μg/μL). Prepared proteins were separated on a 10% polyacrylamide gel via electrophoresis and then transferred to polyvinylidene fluoride (PVDF) membranes using a Mini PROTEIN Tetra Cell System (Bio-Rad, Hercules, CA, USA). The membranes were blocked with 5% skim milk for 1 h, washed once with Tris-buffered saline with 5% Tween 20 (TBST), and incubated with a primary antibody overnight at 4 °C. In the current study, antibodies against inducible nitric oxide synthase (iNOS), cyclooxygenase 2 (COX2), nuclear factor-κB (NF-κB), inhibitor of NF-κB (IκB), phosphorylated IκB, TLR4, and lipopolysaccharide-binding protein (LBP) were used as primary antibodies (Appendix A). Subsequently, the membranes were washed three times with TBST and then incubated with horseradish peroxidase-conjugated IgG for two hours at room temperature. After a final wash with TBST, enhanced chemiluminescence (ECL) reagent (Thermo Fisher Scientific, San Jose, CA, USA) was used to detect the target protein. The membranes were visualized using a FUSION Solo Imaging system (Vilber Lourmat, Marne-la-Vallée, France). The density values were semiquantified using ImageJ (1.52a, National Institutes of Health, Bethesda, MD, USA).

### 4.11. Nuclear Translocation of NF-κB 

To investigate the translocation of NF-κB, two methods (Western blot and immunofluorescence) were used. RAW264.7 cells were seeded in 60 mm cell culture dishes at a density of 1.5 × 10^6^ cells/dish. Cells were incubated with GSE or ginsenosides Rd or Re for 2 h, then treated with LPS (100 ng/mL) for 2 h. After LPS-stimulation, the cells were separated into cytoplasmic and nuclear extracts by using the NE-PER^®^ Nuclear and Cytoplasmic Extraction Reagents kit (Thermo Fisher Scientific, San Jose, CA, USA) for Western blot, or cells were fixed with 4% paraformaldehyde and then stained with primary NF-κB antibody for immunofluorescence analysis.

### 4.12. Statistical Analysis

The data are expressed as the means ± standard deviations (SD) or fold changes in the means. Statistical significance was determined by using one-way analysis of variance (ANOVA) followed by Dunnett’s test. In all analyses, *p* < 0.05 was considered to indicate statistical significance.

## 5. Conclusions

Taken together, we conclude that ginseng sprout extracts have the potential as a pharmaceutical herbal resource to protect or manage sepsis conditions. However, the present results might have some limitations, such as the adaptation of an LPS-induced artificial animal model, and further requirements of defining the corresponding active compounds that produce the effects are needed in the future. 

## Figures and Tables

**Figure 1 ijms-24-01583-f001:**
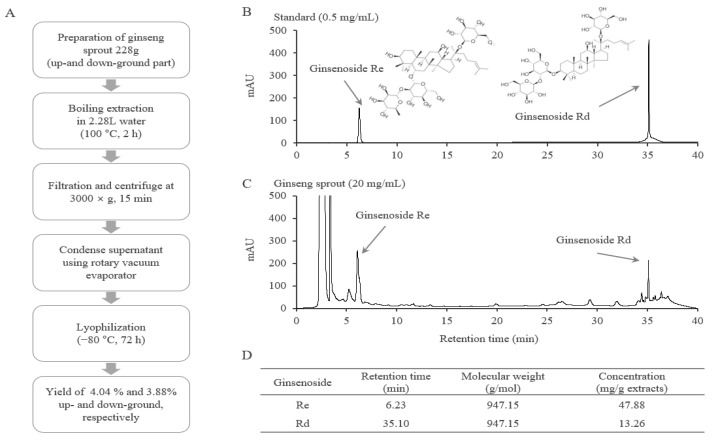
Extract preparation and fingerprinting analysis of GSE. GSE was acquired by water extraction from *Panax ginseng* sprouts (**A**). GSE (**B**) and reference compounds (**C**) were analyzed by high-performance liquid chromatography (HPLC). The concentrations of Re and Rd in the GSE were calculated according to standards (**D**).

**Figure 2 ijms-24-01583-f002:**
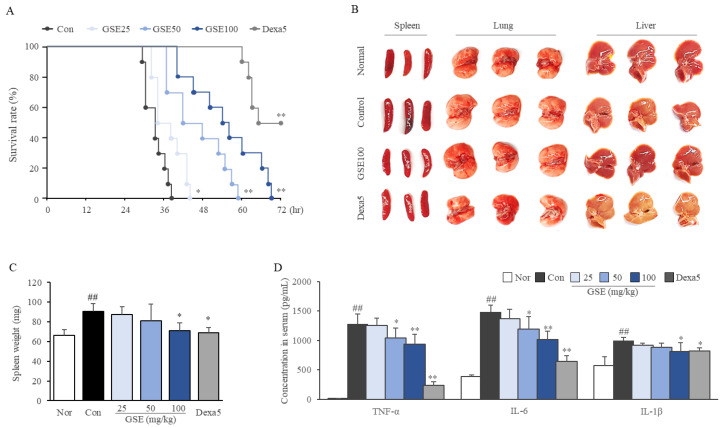
Effects of GSE on survival rate and systemic inflammation. GSE (25, 50, 100 mg/kg) or distilled water was administered by gavage for 5 days (twice per day) to each group of animals (*n* = 15/group). On the final day, all mice were intraperitoneally injected with dexamethasone (5 mg/kg) or normal saline 30 min before LPS injection. After injection of LPS (30 mg/kg), ten mice from each group were monitored to evaluate their survival for 72 h (**A**). Furthermore, 12 h after LPS injection, five mice from each group were sacrificed, and three types of tissues (spleen, lung, and liver) were photographed (**B**). The spleens were weighed (**C**), and the concentrations of serum proinflammatory cytokines (**D**) were determined using ELISAs. Mice or samples were grouped according to treatment: normal (Nor), control (Con), ginseng sprout extract (GSE), and dexamethasone (Dexa5). The data are presented as the mean ± SD. ^##^ *p* < 0.01 compared with the normal group; * *p* < 0.05, ** *p* < 0.01 compared with the control group.

**Figure 3 ijms-24-01583-f003:**
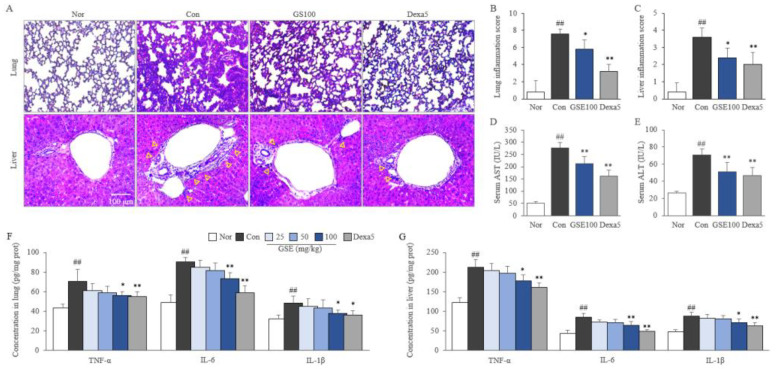
Effects of GSE on pulmonary and hepatic injury. H&E staining (**A**) of lung and liver tissues was performed, and representative photographs (200× magnification) were evaluated for the inflammation scores (**B**,**C**). The yellow triangles indicate the infiltration of immune cells. Serum aspartate aminotransferase (AST) and alanine aminotransferase (ALT) were measured (**D**,**E**). The levels of proinflammatory cytokine in the lung (**F**) and liver (**G**) tissues were analyzed using ELISA. Samples were grouped according to treatment: normal (Nor), control (Con), ginseng sprout extract (GSE), and dexamethasone (Dexa5). The data are presented as the mean ± SD. ^##^ *p* < 0.01 compared with the normal group; * *p* < 0.05, ** *p* < 0.01 compared with the control group.

**Figure 4 ijms-24-01583-f004:**
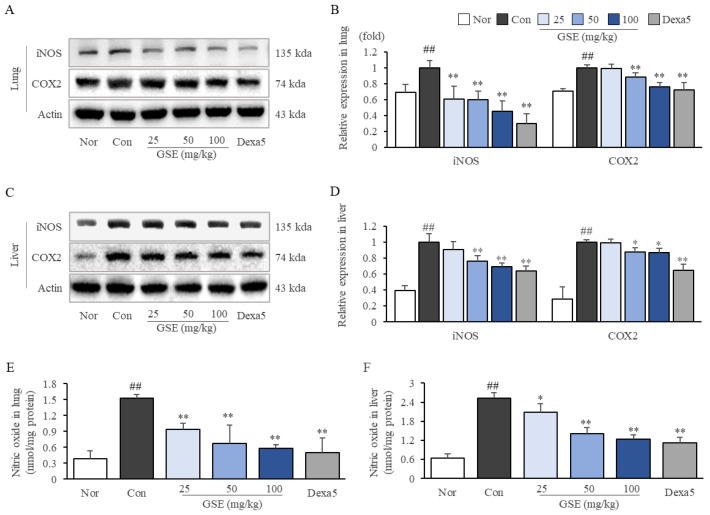
Effects of GSE on inflammatory mediators in pulmonary and hepatic tissues. The protein levels of iNOS and COX2 in the lung (**A**,**B**) and liver (**C**,**D**) tissues were assessed by Western blot, and all the band intensities were quantified by ImageJ. Nitric oxide levels in lung (**E**) and liver (**F**) tissues were evaluated with the Griess reaction. Samples were grouped according to treatment: normal (Nor), control (Con), ginseng sprout extract (GSE), and dexamethasone (Dexa5). The data are presented as the mean ± SD. ^##^ *p* < 0.01 compared with the normal group; * *p* < 0.05, ** *p* < 0.01 compared with the control group.

**Figure 5 ijms-24-01583-f005:**
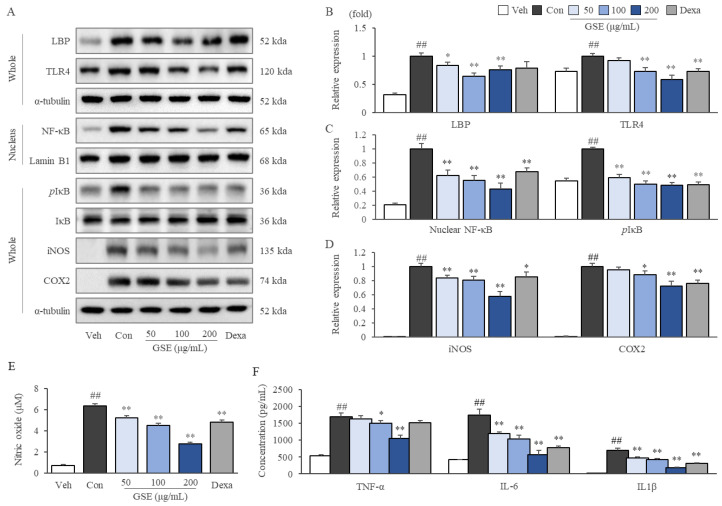
Effects of GSE on the inhibition of TLR4/NF-κB signaling and inflammatory cytokines in LPS-stimulated RAW264.7 cells. Cells were treated with GSE or vehicle for 2 h, after which LPS (100 ng/mL) was added for 2 h (for NF-κB and IκB detection only) or 24 h. Western blot analysis was conducted to evaluate TLR4/NF-κB signaling pathway-associated proteins in LPS-stimulated RAW264.7 cell lysates (**A**), and the intensities of all bands were quantified by ImageJ (**B**–**D**). Cell culture supernatants were used to evaluate nitric oxide (**E**) and proinflammatory cytokines (**F**). Samples were grouped according to treatment: vehicle (Veh), control (Con), ginseng sprout extract (GSE), and dexamethasone (Dexa). The data are presented as the mean ± SD. ^##^ *p* < 0.01 compared with the vehicle group; * *p* < 0.05, ** *p* < 0.01 compared with the control group.

**Figure 6 ijms-24-01583-f006:**
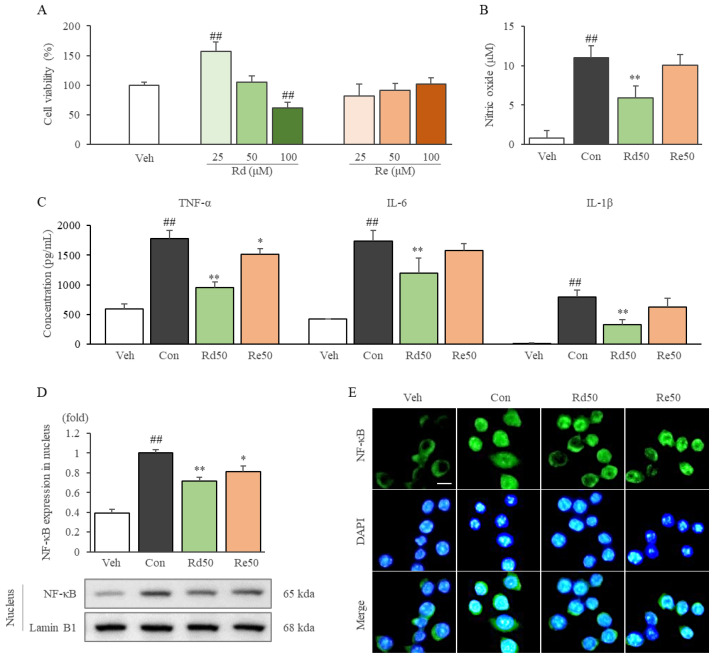
Effects of ginsenoside Rd on the production of inflammatory cytokines and nitric oxide, and nuclear translocation of NF-κB in LPS-stimulated RAW264.7 cells. A WST-8 assay was conducted to evaluate the viability of cells treated with ginsenoside Rd and Re (**A**). Nitric oxide (**B**) and proinflammatory cytokine (**C**) levels were analyzed in the LPS-stimulated-cell supernatant. After 2 h of LPS stimulation, nuclear translocation was evaluated by Western blot (**D**) and immunofluorescence staining under 200× magnification (**E**). Samples were grouped according to treatment: vehicle (Veh), control (Con), 50 μM ginsenoside Rd (Rd50), and 50 μM ginsenoside Re (Re50). The data are presented as the mean ± SD. ^##^ *p* < 0.01 compared with the vehicle group; * *p* < 0.05, ** *p* < 0.01 compared with the control group.

**Table 1 ijms-24-01583-t001:** Cell blood counting analysis.

Hematology Index (10^9^ Cell/L)	Normal	Control	GSE25	GSE50	GSE100	Dexa5
White blood cell	9.98 ± 1.47	4.88 ± 0.29 ^##^	4.30 ± 0.94	5.58 ± 2.19	6.82 ± 1.45 *	5.98 ± 0.77 *
Lymphocyte	7.84 ± 0.92	2.70 ± 0.39 ^##^	2.55 ± 0.64	3.53 ± 0.64	5.02 ± 1.92 *	1.95 ± 1.64 *
Monocyte	0.54 ± 0.05	0.70 ± 0.07 ^##^	0.60 ± 0.07	0.60 ± 0.16	0.54 ± 0.11 *	0.83 ± 0.11
Granulocyte	1.60 ± 0.56	1.48 ± 0.36	1.15 ± 0.23	1.45 ± 0.48	1.26 ± 0.30	3.20 ± 0.39 **

The data are presented as the mean ± SD. ^##^ *p* < 0.01, as compared to normal group; * *p* < 0.05 and ** *p* < 0.01 as compared to Control group. Control group, LPS single treated group; GSE group, LPS with ginseng-sprout-extract-treated group; Dexa5 group, LPS with dexamethasone-treated group.

## Data Availability

The data is contained within the article and the Appendix A.

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
