# Peer review of "Ginseng Sprouts Attenuate Mortality and Systemic Inflammation by Modulating TLR4/NF-κB Signaling in an LPS-Induced Mouse Model of Sepsis"

_ijms, 2023, doi:10.3390/ijms24021583_

Round 1

Reviewer 1 Report

This is a well-designed study employing the in vivo and in vitro assays to characterize the potential role of the ginseng sprout extract (GSE) in antisepsis. The authors provide evidence related to the role of inflammatory cytokines, inflammatory mediators, and the TLR4/NF-kB sigla pathway in GSE’s antisepsis effect, nicely proposing the potential underlying mechanism.

There are a few aspects that need some clarifications:

1.     The authors claim that “Ginsenoside Rd and Re showed nontoxicity up to 50 and 100 μM, respectively” from the viability assay at line 183, however, the bar for Rd at 25µM is labeled as ## (p<0.01) in Figure 6A. Please clarify.

2.     For the same Figure 6A, what is the dosage of Re that starts to show toxic?

3.     Inconsistent descriptions: line 84 mentioned “showed 47.88 mg/g Rd and 13.26 mg/g Re in GSE” and lines 268-269 showed “GSE particularly contained Rd and Re at the as high concentrations of 13.26 and 47.88 mg/g in the extracts”. Which is correct?

4.     The authors have done a similar set of in vitro assays for GSE and Rd/Re in Figure 5 and Figure 6. It is suggested adding a sentence to discuss how much µM of Rd and Re (used in Figure 6) achieves the effect similar to 100 (or 200) mg/mL of GSE (used in Figure 5) in the discussion section.

Author Response

Reviewer #1:
This is a well-designed study employing the in vivo and in vitro assays to characterize the potential role of the ginseng sprout extract (GSE) in antisepsis. The authors provide evidence related to the role of inflammatory cytokines, inflammatory mediators, and the TLR4/NF-kB signal pathway in GSE’s antisepsis effect, nicely proposing the potential underlying mechanism.

  1. The authors claim that “Ginsenoside Rd and Re showed nontoxicity up to 50 and 100 μM, respectively” from the viability assay at line 183, however, the bar for Rd at 25µM is labeled as ## (p < 0.01) in Figure 6A. Please clarify.
    ► Thank you for pointing it out. We confirmed that not only 25 μM but also 12.5 μM of Rd treatment increase cell viability with three repetitions. Although herein the clarified mechanisms of cell viability elevation by low concentration of Rd is difficult to be given, some of the chemicals, such as NAC (N-Acetyl-L-cysteine), also can enhance the cell proliferation under the low concentration due to reduction of doubling time by enhancement of S phase (1). Nevertheless, according to the results of viability, the non-toxicity concentration of Rd (50 μM) was selected for testing pro-inflammatory cytokines. Besides, we described the related contents in the results portion (lines 185-186) and added the repeated cell viability experiment in the supplementary figure 1E.

Reference
1) Mao G, Goswami M, Kalen AL, Goswami PC, Sarsour EH. N-acetyl-L-cysteine increases MnSOD activity and enhances the recruitment of quiescent human fibroblasts to the proliferation cycle during wound healing. Mol Biol Rep. 2016;43(1):31-39.

  1. For the same Figure 6A, what is the dosage of Re that starts to show toxic?
    ► To check the toxic concentration of Re, we additionally performed a cell viability test. Even the high concentrations of Re (250 and 500 μM) also do not show cytotoxicity but markedly increase the cell proliferation (p < 0.05 or 0.01) in RAW264.7 cells. The extra data is added in Supplementary figure 1F.
  2. Inconsistent descriptions: line 84 mentioned “showed 47.88 mg/g Rd and 13.26 mg/g Re in GSE” and lines 268-269 showed “GSE particularly contained Rd and Re at the as high concentrations of 13.26 and 47.88 mg/g in the extracts”. Which is correct?
    ►Sorry for the mistake. GSE contained 13.26 mg/g Rd and 47.88 mg/g Re in the extracts. We corrected the mistake in line 85. As follows, “Semiquantitative analysis showed 47.88 mg/g Re and 13.26 mg/g Rd in GSE.”

  3. The authors have done a similar set of in vitro assays for GSE and Rd/Re in Figure 5 and Figure 6. It is suggested adding a sentence to discuss how much µM of Rd and Re (used in Figure 6) achieves the effect similar to 100 (or 200) mg/mL of GSE (used in Figure 5) in the discussion section.
    ► Thank you for the constructive suggestion. we slightly discussed the related information regarding the dose-effect relationship. As follows:
    Line 274-277: In the present results, similar anti-inflammatory effects were found both in 200 μg/mL of GSE and 50 μM of ginsenoside Rd in LPS-stimulated RAW264.7 cells (Figures 5 and 6). Therefore, the results indicated that the effect of GSE might be due to ginsenoside Rd, not Re.

Reviewer 2 Report

 General comments

In this manuscript, the authors reported their investigation of the effect of ginseng sprout extracts on LPS-induced mouse model of sepsis. The study was carefully designed, the methods were appropriate, the results are interesting and support the conclusion.

Special comments

1.       Abstract. Doses of GSE and LPS, and the number of animals should be included in the abstract.

2.       Line 29. Change “control” to “modulate”.

3.       Lines 31-33. Rephrase.

4.       Line 38. Change “defined” to “as defined”.

5.       Line 52. Delete “On the other hand,”.

6.       Lines 75-76. Rephrase.

7.       Tables and Figures. Please indicate if the values of error or error bars were standard deviation or standard error of the mean.

8.       Some of the bar/column charts used grey scales for different treatments. It would be clearer to use colours.

9.       Line 115. Check spelling “nimals”.

10.   Figure 6A. This showed that the cell viability was increased by 25 uM of ginseng Rd and decreased by 100 uM of ginseng Rd. Could this confound its effect on cytokine production as this might disrupt the cell functions?

11.   Figure 6A. Check the statistical annotation of Re50 as it looks that their mean values were quite close to the Con and their error bars were overlapped.

12.    Line 321. Was there any justification that the mice were given GSE for five days?

13.    Line 327. How did you determine the dose of LPS?

14.    Line 334. What does “every 1 to 72 hours” mean?

15.   Lines 403-404. Did the GSE and LPS incubate with the cells at same time?

16.   Statistical Analysis. Did you check the homogeneity of variance or normality of the data before using parametric test (ANOVA)?

Author Response

Reviewer #2:
In this manuscript, the authors reported their investigation of the effect of ginseng sprout extracts on LPS-induced mouse model of sepsis. The study was carefully designed, the methods were appropriate, the results are interesting and support the conclusion.

  1. Doses of GSE and LPS, and the number of animals should be included in the abstract.
    ► Thank you for pointing it out. we revised the abstract according to the reviewer’s suggestion.
  2. Line 29. Change “control” to modulate
    ► We sincerely appreciate the reviewer’s pointing it out. We changed the term “control” to modulate” according to the reviewer’s suggestion.
  3. Lines 31-33. Rephrase.
    ► We feel thankful for the reviewer’s constructive advice. We realized a strong tone of the sentence is used in the conclusion portion of abstract (Line 31-33). Thus, we rewrote the sentence in Lines 32-34.
  4. Line 38. Change “defined” to “as defined”
    ► Sorry for the mistake. We revised it in Line 39.
  5. Line 52. Delete “On the other hand,”.
    ► Thank you to point it out. We delete it in Line 63
  6. Lines 75-76. Rephrase.
    ► We rewrote the sentence as per to the comment. As shown in lines 73-80.
  7. Tables and Figures. Please indicate if the values of error or error bars were standard deviation or standard error of the mean.
    ►We used the standard deviation to indicate the error bar. Thus, we revised the legends of figures and table.
  8. Some of the bar/column charts used grey scales for different treatments. It would be clearer to use colors.
    ► We sincerely appreciate for reviewer’s advice. We changed the color of the column in the graphs.
  9. Line 115. Check spelling “nimals”.
    ► Sorry for the mistake. We fixed this spelling mistake in line 116.
  10. Figure 6A. This showed that the cell viability was increased by 25 uM of ginseng Rd and decreased by 100 uM of ginseng Rd. Could this confound its effect on cytokine production as this might disrupt the cell functions?
    ► Thank you for the constructive comment. Actually, the purpose of cell viability assay is only for selecting appropriated-concentration applied on further in vitro test. In order to avoid the influence due to proliferation change, we used non-toxic concentration (50 μM of Rd and Re) in next experiments.

Although we did not detect the cytokine levels in Rd 25 μM treatment, massive evidence proved that nitric oxide levels, an essential inflammatory mediator, are positively correlated with pro-inflammatory cytokines (1-2). In the current study, nitric oxide levels in control were dramatically increased (11-fold) as compared to vehicle, and Rd 25 μM (data not shown) and 50 μM treatment showed 20% and 50% decrease respectively in nitric oxide level as compared to the control. Therefore, the pro-inflammatory cytokines might be mildly reduced by Rd 25 μM. Accordingly, we supposed that increase in cell viability by 25uM Rd do not change the function with high possibility.

  1. Figure 6C. Check the statistical annotation of Re50 as it looks that their mean values were quite close to the Con and their error bars were overlapped.
    ► Thank you for pointing it out. We re-analyzed the data, and recognized that there was no significant difference in IL-6 concentration in the Re50 treatment compared to the control. We revised it in Figure 6C.
  2. Line 321. Was there any justification that the mice were given GSE for five days?
    ► Thank you for the reviewer’s meaningful question. Panax ginseng has been commonly used as a nourishing herb to strengthen the body and prevent various diseases for a long time. In the clinic, at least 5 days as a course of ginseng treatment has been recommended by oriental doctors. Due to the above reason, we pretreated GSE for five days in the present study.
  3. Line 327. How did you determine the dose of LPS?
    ► We sincerely appreciate for reviewer’s questions. The dose selection in the present study is based on previously published articles and our pilot study. Various articles selected 30 or 40 mg/kg of LPS to induce sepsis (1, 2) Based on these references, we performed the pilot experiments in 30, 40, 60, and 75 mg/kg of LPS injection. 40 mg/kg of LPS killed all mice up to 24 h, but 30 mg/kg of LPS injection allowed them to live up to 38 hours. In both doses of LPS, GSE could attenuate septic death time (Figure. 2A and Supplementary figure 2). We added the supplementary figure description in the manuscript (Line 332-333).

    Reference
    1) Paik, S., Choe, J.H., Choi, G.-E., Kim, J.-E., Kim, J.-M., Song, G.Y., and Jo, E.-K. (2019). Rg6, a rare ginsenoside, inhibits systemic inflammation through the induction of interleukin-10 and microRNA-146a. Scientific reports 9, 1-15.
    2) Yang, C.S., Ko, S.R., Cho, B.G., Shin, D.M., Yuk, J.M., Li, S., Kim, J.M., Evans, R.M., Jung, J.S., and Song, D.K. (2008). The ginsenoside metabolite compound K, a novel agonist of glucocorticoid receptor, induces tolerance to endotoxin‐induced lethal shock. Journal of cellular and molecular medicine 12, 1739-1753.
  4. Line 334. What does “every 1 to 72 hours” mean?
    ► Sorry for the description mistake. We did not express the meaning properly in this sentence. We paraphrase the sentence like “The survival rate was recorded at 1-hour intervals from the start time to 72 hours.” in lines 339-340.
  5. Lines 403-404. Did the GSE and LPS incubate with the cells at same time?
    ► Sorry for the description error. We pre-treat with GSE or ginsenosides for 2 hours. And then, 100 ng/mL of LPS was added to the cell medium (including GSE or ginsenosides). Thus, we revised the explanation to “Cells were incubated with GSE or ginsenosides Rd or Re for 2 hours, and then treated with LPS (100 ng/mL) for 2 hours.” in lines 408-409.

Statistical Analysis. Did you check the homogeneity of variance or normality of the data before using parametric test (ANOVA)?
► Thanks a lot for the professional comment. In the study, the Shapiro-Wilks test was performed to check all data for normality by SPSS 20.0, and then the ANOVA test (post-hoc test of Dunnett) was performed by prism 7.0.

Reviewer 3 Report

Although the technologies and results sound well, novelty of the study was low. Many reprots had showed the anti-inflammatory effects of Ginseng extracts and ginsenosides on LPS-induced lung injury and sepsis. And also the some targets were presented as previously reported.

Author Response

Reviewer #3:

  1. Although the technologies and results sound well, novelty of the study was low. Many reports had showed the anti-inflammatory effects of Ginseng extracts and ginsenosides on LPS-induced lung injury and sepsis. And also some targets were presented as previously reported.
    ► We fully understand the reviewer’s opinion.
    Firstly, although matured ginseng (6-year root) has been well studied on many pharmaceutic properties, so far no study has been performed regarding ginseng sprout against sepsis, a severe disease with high mortality.
    Secondly, ginseng sprout is different from matured ginseng in several aspects. Whole ginseng sprout has more obvious advantages, e.g. very low price (approximately 1/10), short harvest period (2-8 weeks), and high in several Ginsenosides (Rd, Re, etc.).
    Eventually, we supposed that the GSE has enough novelty both in the academic and commercial fields.

Round 2

Reviewer 3 Report

In folk medicine, the matured ginseng was most widely used in clinic rather than the sprout. If the sprout had more anti-inflammatory effects against LPS-induced sepsis, the authors should present the chemical data on ginsenoside contents and therapeutic difference between the matured and the sprout. The conclusion of study seemed lack of evidence when the comparation was miss.

Author Response

  1. In folk medicine, the matured ginseng was most widely used in clinic rather than the sprout. If the sprout had more anti-inflammatory effects against LPS-induced sepsis, the authors should present the chemical data on ginsenoside contents and therapeutic difference between the matured and the sprout. The conclusion of study seemed lack of evidence when the comparation was miss.
    ► We fully understand the reviewer’s opinion.
    It is true that six-year-old ginseng is used most clinically, but it is also true that ginseng sprout is emerging recently. However, despite the rapid cultivation and high ginsenoside content of ginseng sprout, no studies have yet revealed its effectiveness on sepsis. So we wanted to make this clear, and as the reviewer said, we're going to do another study in the future to compare the effects of ginseng sprout and matured ginseng.
